# Cancer cluster among small village residents near the fertilizer plant in Korea

**Hyungryul Lim** [1], **Yong-Han Lee**[1], **Sanghyuk Bae**[2], **Do-Hyun Koh** [3], **Mira Yoon**[4], **Bo-Eun Lee**[4], **Jeong-Soo Kim**[5]☯*, **Ho-Jang Kwon** [1]☯*

1 Department of Preventive Medicine, College of Medicine, Dankook University, Cheonan, Chungcheongnam-do, Republic of Korea, 2 Department of Preventive Medicine, College of Medicine, Catholic University of Korea, Seoul, Republic of Korea, 3 Citizen Science Institute, Goyang, Gyeonggi-do, Republic of Korea, 4 Environmental Research Department, National Institute of Environmental Research, Incheon, Republic of Korea, 5 Institute for Environmental Safety and Health, Seoul, Republic of Korea

☯ These authors contributed equally to this work.
* eco@ecosafety.kr (JSK); hojangkwon@gmail.com (HJK)

## Abstract

### Objectives

In Jang-jeom, a small village in Hamra-myeon, Iksan-si, Jeollabuk-do, South Korea, residents raised concerns about a suspected cancer cluster that they attributed to a fertilizer plant near the village. We aimed to investigate whether the cancer incidence in the village was higher than that in the general Korean population when the factory was in operation (2001–2017) and whether living in the village was associated with a higher risk of cancer.

### Methods

Using national population data and cancer registration data of South Korea, we estimated the standardized incidence ratios (SIRs) in the village to investigate whether more cancer cases occurred in the village compared to other regions. The SIRs were standardized by age groups of 5 years and sex. In order to investigate whether residence in the village increased the risk of cancer, a retrospective cohort was constructed using National Health Insurance Service (NHIS) databases. We estimated the cancer hazard ratios (HRs) using the Cox proportional hazard model, and defined the exposed area as the village of Jang-jeom, and the unexposed or control area as the village neighborhood in Hamra-myeon. We considered potential confounding variables such as age, sex, and income index in the models. Additionally, we measured polycyclic aromatic hydrocarbons (PAHs) and tobacco-specific nitrosamines (TSNAs), suspected carcinogens that may have caused the cancer cluster, in samples collected from the plant and the village.

### Results

Twenty-three cancer cases occurred in Jang-jeom from 2001 to 2017. Between 2010 and 2016, the incidence rates of all cancers (SIR: 2.05, except thyroid cancer: 2.22), non-melanoma skin cancer (SIR: 21.14, female: 25.41), and gallbladder (GB) and biliary tract cancer in men (SIR: 16.01) in the village were higher than those in the national population in a way

**Data Availability Statement:** Data are available from Korean National Health Insurance Sharing Service (NHISS). However, data cannot be shared publicly because KNHISS does not allow researchers to provide data personally or share

publicly. However, all researchers can access the data if applying in the same manner as the authors upon completing the online data request form. Access to NHIS-customized data can be achieved from the website of NHISS (https://nhiss.nhis.or.kr/) after promising to follow the research ethics and completing the application process (https://nhiss.nhis.or.kr/bd/ab/bdaba032eng.do). If approved, the researchers can receive and analyze the data only at NHIS Big Data Analysis Center. The authors did not have any special access privileges to the database.

**Funding:** This work was supported by a grant from the National Institute of Environment Research (NIER), funded by the Ministry of Environment (MOE) of the Republic of Korea (NIER-2017-03-02-053).

**Competing interests:** The authors declare they have no actual or potential conflicts of interest associated with the material presented in this paper.

that was statistically significant. In our cohort analysis that included only Hamra-myeon residents who have lived there for more than 7 years, we found a statistically significant increase in the risk of all cancers (HR: 1.99, except thyroid cancer: 2.20), non-melanoma skin cancer (HR: 11.60), GB and biliary tract cancer (HR: 15.24), liver cancer (HR: 6.63), and gastric cancer (HR: 3.29) for Jang-jeom residents compared to other Hamra area residents. We identified PAHs and TSNAs in samples of deposited dust and residual fertilizer from the plant and TSNAs in dust samples from village houses.

## Conclusions

The results of the SIR calculation and cancer risk analyses of Jang-jeom village residents from the retrospective cohort design showed consistency in the effect size and direction, suggesting that there was a cancer cluster in Jang-jeom. This study would be a good precedent for cancer cluster investigation.

## Introduction

In April 2017, residents of Jang-jeom in Hamra-myeon, Iksan-si, Jeollabuk-do, petitioned the South Korean government for an epidemiological investigation, claiming that out of a total of less than 100 villagers, 19 people had developed cancer and 10 of them had died since 2010. The village people blamed the stench and waste water coming out of the total waste recycling fertilizer production plant, which was located 500 meters from the village entrance, for the cancer occurrences. At that time, it was revealed that the company had been illegally using tobacco leaves to produce fertilizers. In April 2017, the plant was ordered to be closed down for emitting pollutants without permission from Iksan-si and was eventually shut down [1].

The Korean government accepted the petition, and multi-disciplinary experts started conducting an epidemiological investigation to scientifically evaluate whether a real cancer cluster outbreak has occurred in the village. If it turned out to be a real cluster, they would have to identify whether it was linked to pollutant exposure from the fertilizer plant.

A cancer cluster can be defined as a higher-than-expected incidence of cancer in a specific group in a particular geographical space over a period of time [2]. Therefore, in this study, we used statistical methods to primarily investigate whether the cancer incidence in the village was higher than that of the general Korean population when the factory was in operation (2001–2017) and whether living in the village was associated with a higher cancer risk.

## Materials and methods

### Collecting national data

In early 2018, when the study began, the plant had already closed down, and many of the cancer patients had passed away. Therefore, carrying out a direct survey was not feasible. In order to determine whether there was a cancer cluster in Jang-jeom village, the study needed to reconstruct the past with more objective data. The sources and purpose of the data collecting are shown in Fig 1. We collected Hamra-myeon office registry data to identify the actual number of village residents by year, Korean Statistical Information Service (KOSIS) data to estimate the standard population size using statistical calculations, Korean Central Cancer Registry (KCCR) data to identify the actual number of cancer cases in the village and across Korea, and the National Health Insurance Service's (NHIS) eligibility of the insured and medical treatment data to construct a retrospective cohort [3].

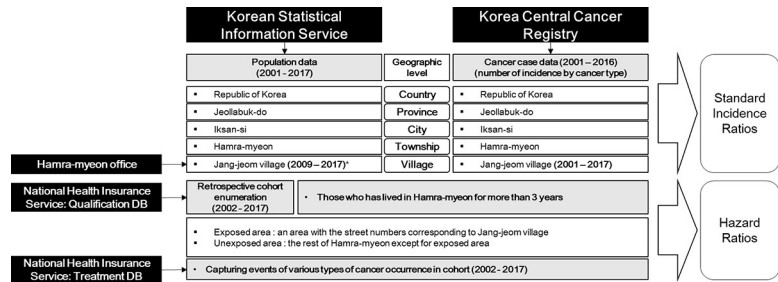

**Fig 1. Sources and purpose of data collected for estimating cancer risks.** Black boxes indicate the data sources for the study, and light gray boxes indicate collected data or database. Final outcome estimates are standard incidence ratios and hazard ratios for cancer in the Jang-jeom village residents. * With an assumption that the Jang-jeom village's demographics for 2001–8 are the same as for 2009, standard incidence ratios for 2001–9 can also be calculated.

### Identifying cancer cases and estimating standardized incidence ratios

We confirmed 31 cancer cases in the village by verifying whether they were registered in KCCR after obtaining consent for the use of their registration numbers. When the cancer registration data assigned the International Classification of Diseases, 10th revision (ICD-10) code of "C00-C97," classified as a malignant neoplasm, it was deemed to be a confirmed case of cancer. Through this process, we identified the cancer cases in the village from 2001, when the plant started operating, to 2017, the most recent year for which data is available in KCCR.

We estimated the standardized incidence ratios (SIRs) in the village to investigate whether more cancer cases occurred in the village compared to other regions. To estimate the SIRs, we needed not only the number of cancer cases, but also the comparative population sizes of the whole of South Korea, Jeollabuk-do, and Iksan-si for the 2001 to 2017 period from the KOSIS website. There was no data on the village population in KOSIS, so we accessed that data for 2009 to 2017 from the Hamra-myeon Office's registry. From the obtained data, we estimated the mid-year population numbers for each year from 2010 to 2017. Since we wanted to estimate SIRs from 2001 to 2009, we adopted the village population structure from 2001 to 2008 by deducting 5 years from the demographic structure of 2009.

We estimated SIRs adjusted for age group of 5 years and sex for all malignant cancers and specific types of cancer that had been confirmed among the residents from 2001 to 2016. We set the whole of South Korea, Jeollabuk-do, Iksan-si, and Hamra-myeon as the standard populations to estimate SIRs for each area. SIRs were presented for both periods (2001 to 2009 and 2010 to 2016), and cumulative estimates were given from 2001 to each year. In terms of the number of cancer cases in the whole of Korea, Jeollabuk-do, Iksan-si, and Hamra-myeon for comparison, we only assessed the data from 2001 to 2016 from KCCR (Table 1) that was why we could not estimate the SIR of 2017. Point estimates and 95% confidence intervals (CIs) of each SIR were derived from the following formula [4]:

$$Standardized\ Incidence\ Ratio\ (SIR) = \sum_{k=1}^{M} O_k \bigg/ \sum_{k=1}^{M} t_k \lambda_k = O/E$$

$$95\%\ Confidence\ Interval\ (CI) of\ SIR = [\sqrt{O} \pm (Z_{0.05/2} \times 0.5)]^2 / E$$

where M cells are defined by the cross-classification of gender (male and female) and age group (5-year unit) of the cohort, $O_k$ represents the observed events in the cohort subjects contributing to the $k^{th}$ cell, $t_k$ represents the total person-time of the cohort in the $k^{th}$ cell, $\lambda_k$ represents the standard rate corresponding to the $k^{th}$ cell, where $k = 1, 2, \ldots, M$, $O$ represents the

**Table 1. The cancer standardized incidence ratio of the Jang-jeom village compared to the whole of Korea in 2010–16.**

| | Sex | Jang-jeom's 2013 population | Observed cases | Expected cases | SIR | (95% CI) |
|---|---|---|---|---|---|---|
| All cancers (C00-96) | Men | 58 | 7 | 3.76 | 1.86 | (0.74–3.49) |
| | Women | 52 | 6 | 2.71 | 2.22 | (0.80–4.34) |
| | Total | 110 | 13 | 6.35 | 2.05 | (1.09–3.31) |
| All cancers except thyroid cancer (C00-72, 74–96) | Men | 58 | 7 | 3.67 | 1.91 | (0.76–3.59) |
| | Women | 52 | 6 | 2.31 | 2.60 | (0.93–5.09) |
| | Total | 110 | 13 | 5.85 | 2.22 | (1.18–3.59) |
| Hepatic cancer (C22) | Men | 58 | 1 | 0.34 | 2.95 | (0.00–11.56) |
| | Women | 52 | 0 | 0.15 | | - |
| | Total | 110 | 1 | 0.48 | 2.10 | (0.00–8.21) |
| Skin cancer except melanoma (C44) | Men | 58 | 1 | 0.07 | 14.11 | (0.01–55.33) |
| | Women | 52 | 3 | 0.12 | 25.41 | (4.79–62.30) |
| | Total | 110 | 4 | 0.19 | 21.14 | (5.50–46.93) |
| Gallbladder and biliary cancer (C23-4) | Men | 58 | 2 | 0.12 | 16.01 | (1.51–45.88) |
| | Women | 52 | 0 | 0.13 | | - |
| | Total | 110 | 2 | 0.25 | 8.07 | (0.76–23.13) |
| Gastric cancer (C16) | Men | 58 | 2 | 0.65 | 3.06 | (0.29–8.78) |
| | Women | 52 | 1 | 0.33 | 3.06 | (0.00–11.99) |
| | Total | 110 | 3 | 0.96 | 3.13 | (0.59–7.68) |
| Breast cancer (C50) | Men | 58 | 0 | 0.00 | | - |
| | Women | 52 | 1 | 0.28 | 3.58 | (0.00–14.03) |
| | Total | 110 | 1 | 0.29 | 3.50 | (0.00–13.71) |
| Lung cancer (C33-4) | Men | 58 | 1 | 0.68 | 1.47 | (0.00–5.77) |
| | Women | 52 | 1 | 0.28 | 3.63 | (0.00–14.24) |
| | Total | 110 | 2 | 0.92 | 2.18 | (0.21–6.25) |

SIR: Standardized incidence ratio. CI: Confidence interval. -: If there were no cases in the Jang-jeom village in the period, the calculation is impossible and the SIRs and 95% CIs are marked with "-".

total number of observed events, and $E$ represents the total number of expected events. $Z_{0.05/2}$ represents a z-score of 0.025.

## Estimating cancer hazard ratios from retrospective cohort design

To determine if the risk of cancer increased due to residence in Jang-jeom, a cohort-based study design and relevant statistical modeling were required. In this study, a retrospective cohort was constructed using the NHIS databases. In South Korea, the whole population is enrolled in one central insurer, NHIS. Since NHIS enrollment is mandatory, through the insurance system, we could obtain the qualification database of the service (income level, resident area, etc.) and medical treatment databases (date of visit, ICD code, prescription, etc.) of all Hamra-myeon residents including Jang-jeom villagers and reconstruct a detailed cohort at an individual level [3].

We defined residence history in Jang-jeom village as "exposure" and residence history in the rest of the Hamra-myeon area except the village as "non-exposure" or "control" from 2002, the earliest year for which NHIS data was available, to 2017. The entry date was set to January 1 of the year if the residence code for each year after 2002 was Hamra-myeon, and the end of the follow-up date was set to December 31 of the year before the year in which the code was changed to an area other than Hamra-myeon. If someone whose follow-up had ended moved

again in Hamra-myeon, they were included in the cohort from January 1 of the year. The reason we reconstructed as described above is that the residence codes are updated in 1-year units for migration of the insured in the NHIS databases. Those who had lived in both the exposed and control areas were excluded from the cohort because they were not suitable for determining the association between area of residence and cancer incidence.

A cancer case, an event of the cohort, was classified and analyzed using the ICD-10 code. We defined "C00-96" as all cancers, "C00-72, 74–96" as all cancers except thyroid cancer, "C22" as hepatic cancer, "C44" as skin cancer except melanoma, "C23-4" as gallbladder (GB) and biliary cancer, "C16" as gastric cancer, "C18-20" as colorectal cancer, "C50" as breast cancer, "C25" as pancreatic cancer, and "C33-4" as lung cancer.

The date of occurrence of cancer was designated as the earliest date of the first treatment based on medical treatment bills in medicine, dentistry, and pharmacy, but not Korean medicine. Some cases might be missed by applying only the main disease code (the first disease code), so the sub-disease code (the second disease code) was also applied to calculate the number of cases. However, this could lead to the identification of false cases, so only those who had medical treatment bills with cancer-related exemption calculation codes (V027, V193, and V194) were defined as cancer cases as these codes clarified whether the subject had received cancer treatment. The exemption calculation system is an arrangement where the out-of-pocket expenses for insured medical care is reduced to 5% in cancer cases (as well as other severe diseases), which means that the presence of these codes in bills reaffirms the disease code for cancer in the NHIS databases.

It takes several years for environmental exposure to be enough to cause cancer in humans. Therefore, it was necessary to exclude from our analyses people whose residence period was shorter than at least 3 years. We constructed three cohorts with minimum residence periods of 3 years, 5 years, and 7 years to see whether there was consistency or any pattern in the results of each cohort. We applied the Cox proportional hazard model to estimate the hazard ratios (HRs) and 95% CIs for the village residents compared to those in the control area. To adjust for potential confounders, individual-level information on age, sex, and income level from the NHIS eligibility of the insured database was included in the adjusted model as covariates. For the equalization of income, the Organization for Economic Cooperation and Development's square root scale (household income/$\sqrt{}$household size) was used. All analyses used R software (version 3.5.1, R Project for Statistical Computing, Vienna, Austria), and the statistical significance level ($\alpha$) was set at 0.05. Additional R packages used for analyses were "survival" and "survminer."

## Measurement of polycyclic aromatic hydrocarbons and tobacco-specific nitrosamines in collected samples

First, to check how much tobacco leaves were actually used at the fertilizer plant, we examined the data in the Korean Ministry of Environment's Allbaro online waste management system. The nationwide system monitors the entire process from the discharge of waste at a plant to transportation and final treatment on a real-time basis in a transparent manner.

We decided to measure polycyclic aromatic hydrocarbons (PAHs) and tobacco-specific nitrosamines (TSNAs), which are known human carcinogens generated from the processing of tobacco leaves [5], in samples of deposited dust, wastewater, and residual fertilizer collected from the plant, and in the raw tobacco leaf material kept in the village. We obtained 18 samples of deposited dust (8 from the wall, 4 from the floor, 5 around the facility, and 1 from the ceiling frame). Two wastewater samples were collected from the wastewater collection tank and the raw material mixing section at the plant using a water collection machine, while two samples of residual fertilizer, one from inside the stirrer and the other from inside the dryer, were also obtained.

Further, we investigated whether the pollutants emitted from the plant affected the village. Two control areas, Ganseong village and Topgoji village, were selected for comparison based on the criteria that they were also located in Hamra-myeon, but not affected by the pollutants, and that they had a similar population as Jang-jeom. A map of the study area is shown in Fig 2. From 15 sites in Jang-jeom and 5 sites in the control villages, we obtained 50g samples of dust deposited on the rooftops and roofs of houses. PAHs and TSNAs were measured in the samples. We used a gas chromatography with tandem mass spectrometry (GC-MS/MS) instrument manufactured by Agilent Technologies (CA, USA) to quantify PAHs and a liquid chromatography with tandem mass spectrometry (LC-MS/MS) instrument manufactured by Agilent to quantify TSNAs in our samples. The study quantified the four TSNAs: N-nitrosonornicotine (NNN), 4-(methylnitrosamino)-1-(3-pyridyl)-1-butanone (NNK), N-nitrosoanatabine (NAT), and N-nitrosoanabasine (NAB). More detailed sample analysis procedure is presented in S1 Appendix.

## Ethics committee approval and consent of participants

The research process of this study was approved by the Institutional Review Board (IRB) of Wonkwang University Hospital (WKUH-2018-04-017). Cancer identification in Jang-jeom through KCCR was proceeded with individual consent for collection and use of national identification numbers. In terms of using NHIS database, the consent was waived by the IRB under the terms of use of anonymized secondary data with public purpose.

## Results

### Identifying cancer cases and estimating SIRs

From 2001 to 2017, the total number of cancer patients in Jang-jeom village was 22, with one resident suffering from two types of cancer, resulting in a total of 23 cancer cases. The cases

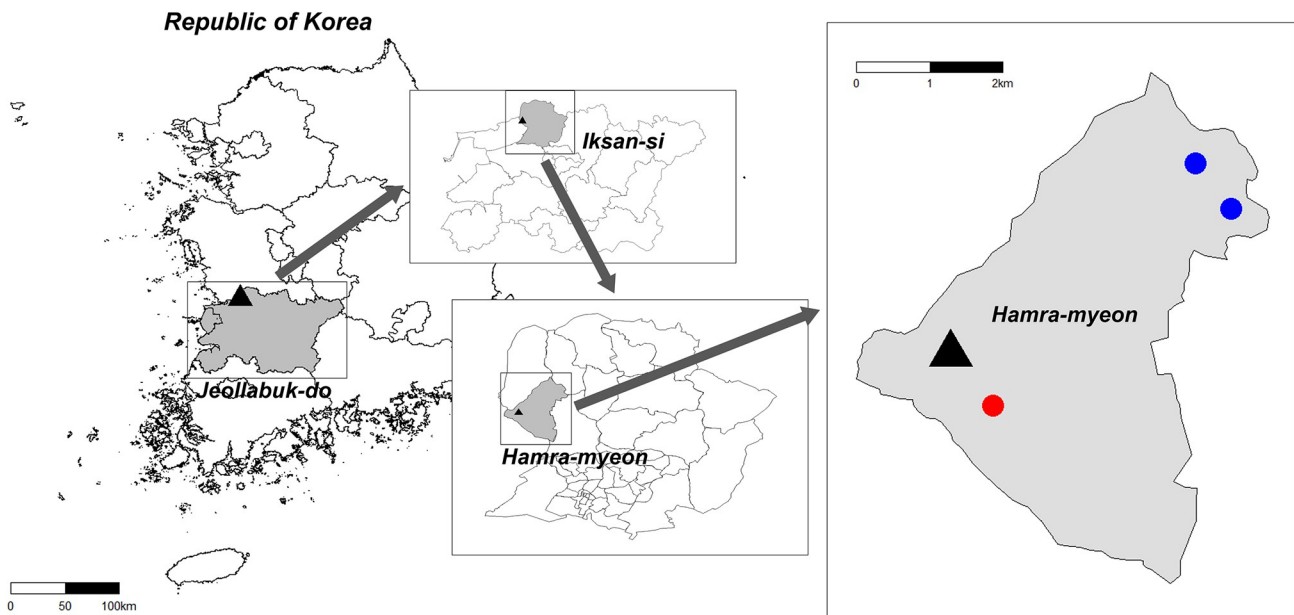

**Fig 2. The map of study area, Hamra-myeon, Iksan-si, Jeollabuk-do, South Korea.** Jeollabuk-do is a province located in the southwest of South Korea. Iksan-si is a city located in the northwest of Jeollabuk-do. Hamra-myeon is one of sub-administrative districts of Iksan-si, located in the west of the city. The black triangle indicates the fertilizer factory near Jang-jeom village marked with red dot, and two blue dots indicates Ganseong (upper) and Topgoji villages (lower), which are also located in Hamra-myeon, but thought to be not affected by the pollutants emitted from the plant.

included 1 hepatic cancer, 4 non-melanoma skin cancers, 3 GB and biliary cancers, 3 colon cancers, 6 gastric cancers, 1 breast cancer, 1 pancreatic cancer, and 4 lung cancers (Fig 3).

Table 1 shows the SIRs for the village in comparison with the values for the whole of Korea from 2010 to 2016. For both men and women, the SIRs of 2.05 (95% CI: 1.09–3.31) for all cancers, 2.22 (95% CI: 1.18–3.59) for all cancers except thyroid cancer, and 21.14 (95% CI: 5.50–46.93) for non-melanoma skin cancer were statistically significant. In the case of non-melanoma skin cancer, the SIR was especially high for women (25.41, 95% CI: 4.79–62.30). In contrast, the SIR of GB and biliary cancer for both men and women was not statistically significant (8.07, 95% CI: 0.76–23.13), but it was higher in men (16.01, 95% CI: 1.51–45.88). We also estimated SIRs using the standard population of Jeollabuk-do, Iksan-si, and Hamra-myeon from 2010 to 2016. The values were comparable to the nationwide estimates. The SIRs from 2010 to 2016 were higher than those from 2001 to 2009 in most cases where comparison was possible (S1 Table).

The cumulative SIRs for the village for each year from 2001 to 2016 compared to the whole of Korea for all cancers (except thyroid cancer), non-melanoma skin cancer, and GB and biliary cancer are shown in Fig 2. In 2001, when the analysis began, there was 1 colon cancer case in the village, so the cumulative SIRs for the first one or two years were high, but the cumulative SIRs for all cancers have been increasing in recent years (Fig 4A). Since there were no thyroid cancer cases in the village, the overall SIRs for all cancers except thyroid cancer were higher than those of all cancers (Fig 4B). The cumulative SIRs of non-melanoma skin cancer have increased since 2012, when the first case was identified (Fig 4C). In addition, the SIRs of GB and biliary cancer have also increased since 2008, when the first case was identified. Since 2012, the SIRs have been getting lower as new cases have not occurred (Fig 4D).

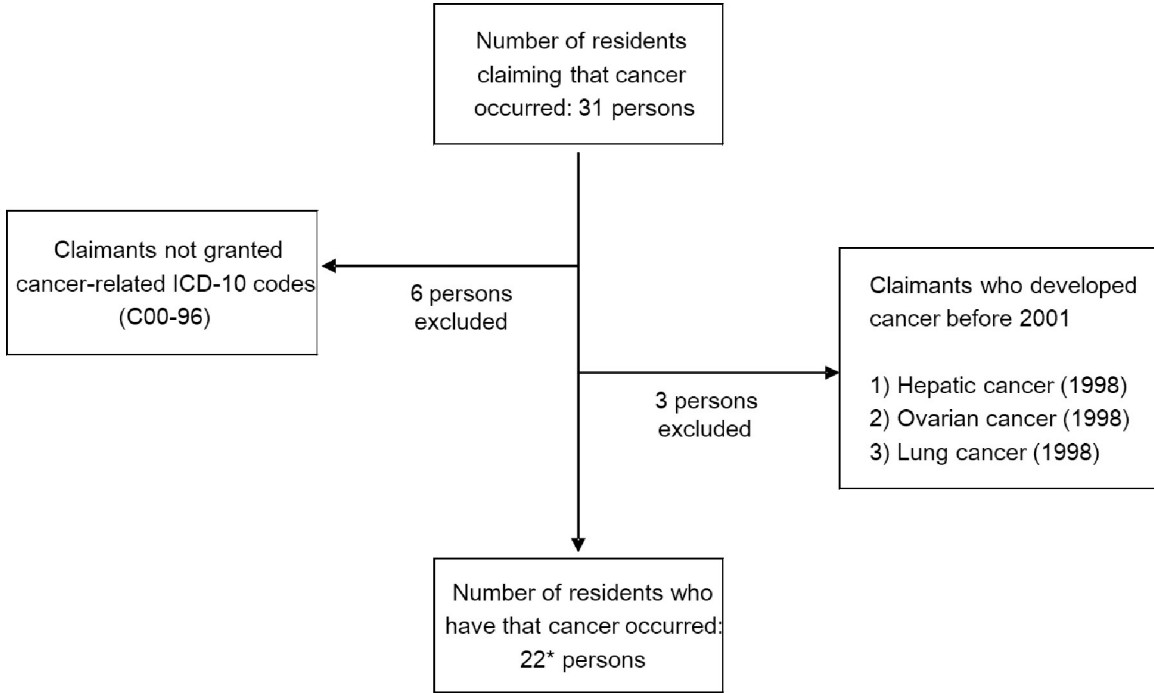

**Fig 3. Validation of cancer cases in the Jang-jeom village in 2001–17.** *: There was one person with two cancers (colon cancer, gallbladder and biliary cancer), so the total number of cases was 23.

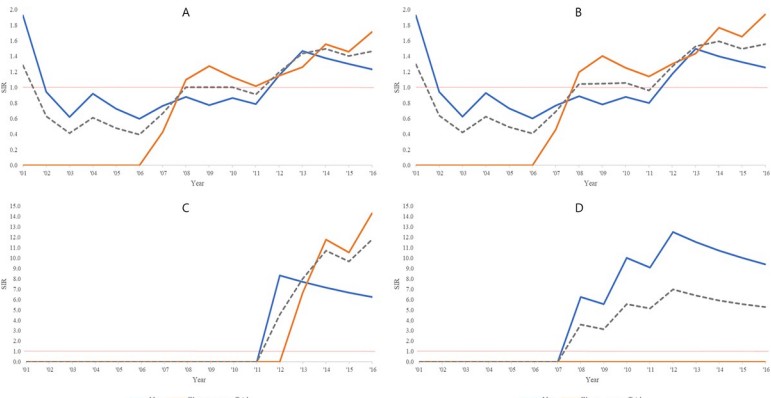

**Fig 4. The cumulative cancer standardized incidence ratio by year of the Jang-jeom village compared to the whole of Korea in 2010–16.** A) All cancers (C00-96), B) All cancers except thyroid cancer (C00-72, 74–96), C) Skin cancer except melanoma (C44), and D) Gallbladder and biliary cancer (C23, C24).

## Estimating cancer HRs

The demographic characteristics of the cohort subjects are shown in Table 2. In each cohort, there were more men in the village group than in the control group, and the average age was higher and the average income index was lower in the village group than in the control group.

In the Cox proportional hazard model adjusted for age, sex, and income level, the cancer incidence was higher in a statistically significant way in the village group than in the control group for more than 7 years for all cancers (HR = 1.99, 95% CI: 1.14–3.45), all cancers except thyroid cancer (HR = 2.20, 95% CI: 1.26–3.84), hepatic cancer (HR = 6.63, 95% CI: 1.77–24.88), non-melanoma skin cancer (HR = 11.60, 95% CI: 2.95–45.54), GB and biliary cancer (HR = 15.24, 95% CI: 3.84–60.48), and gastric cancer (HR = 3.29, 95% CI: 1.14–9.53). In all three models with minimum residence periods of 3 years, 5 years, and 7 years, the HRs for all cancers except thyroid cancer, GB and biliary cancer, and non-melanoma skin cancer were higher in a way that was statistically significant. Overall, the longer the minimum residence period in the model, the higher the HR point estimates (Table 3). The results of the unadjusted models are presented in S2 Table.

**Table 2. Characteristics of retrospective cohort subjects according to minimum observation period.**

| | Minimum observation period: 3 years | | | Minimum observation period: 5 years | | | Minimum observation period: 7 years | | |
|---|---|---|---|---|---|---|---|---|---|
| | Total | Jang-jeom village | Control area* | Total | Jang-jeom village | Control area* | Total | Jang-jeom village | Control area* |
| | N = 4,400 | N = 160 | N = 4,240 | N = 3,258 | N = 134 | N = 3,124 | N = 2,654 | N = 119 | N = 2,535 |
| Sex, N (%) | | | | | | | | | |
| Men | 2,226 (50.6) | 85 (53.1) | 2,141 (50.5) | 1,670 (51.3) | 74 (55.2) | 1,596 (51.1) | 1,341 (50.5) | 62 (52.1) | 1,279 (50.5) |
| Women | 2,174 (49.4) | 75 (46.9) | 2,099 (49.5) | 1,588 (48.7) | 60 (44.8) | 1,528 (48.9) | 1,313 (49.5) | 57 (47.9) | 1,256 (49.5) |
| Age (year), Mean (SD) | 39.6 (22.4) | 42.2 (23.4) | 39.5 (22.3) | 41.8 (22.2) | 44.0 (23.3) | 41.7 (22.2) | 42.5 (22.1) | 44.0 (23.2) | 42.4 (22.1) |
| Income index†, Mean (SD) | 25,320 (23,169) | 18,572 (16,061) | 25,579 (23,362) | 24,925 (23,227) | 18,239 (15,542) | 25,218 (23,464) | 22,410 (16,343) | 18,286 (15,633) | 22,608 (16,353) |

*: The rest of the Hamra-myeon except the Jang-jeom village. †: National health insurance payment amount/√number of households.

**Table 3. The hazard ratio of the living in the Jang-jeom village compared to the neighborhood area.**

| | Minimum observation period: 7 years | | Minimum observation period: 5 years | | Minimum observation period: 3 years | |
|---|---|---|---|---|---|---|
| | HR* (95% CI) | | HR* (95% CI) | | HR* (95% CI) | |
| All cancers (C00-96) | 1.99 | (1.14–3.45) | 1.82 | (1.10–3.01) | 1.57 | (0.99–2.49) |
| All cancers except thyroid cancer (C00-72, 74–96) | 2.20 | (1.26–3.84) | 1.97 | (1.19–3.25) | 1.61 | (1.01–2.58) |
| Hepatic cancer (C22) | 6.63 | (1.77–24.88) | 3.59 | (1.03–12.47) | 2.13 | (0.64–7.05) |
| Thyroid cancer (C73) | - | | - | | 0.87 | (0.12–6.47) |
| Skin cancer except melanoma (C44) | 11.60 | (2.95–45.54) | 9.56 | (2.68–34.09) | 9.82 | (2.73–35.38) |
| Gallbladder and biliary cancer (C23-4) | 15.24 | (3.84–60.48) | 10.40 | (3.41–31.69) | 7.58 | (2.65–21.64) |
| Colorectal cancer (C18-20) | 0.65 | (0.09–4.79) | 0.58 | (0.08–4.29) | 0.74 | (0.18–3.07) |
| Gastric cancer (C16) | 3.29 | (1.14–9.53) | 2.14 | (0.76–6.05) | 1.70 | (0.61–4.77) |
| Breast cancer (C50) | - | | - | | 2.11 | (0.26–16.88) |
| Pancreatic cancer (C25) | - | | 2.41 | (0.29–19.75) | 1.97 | (0.25–15.81) |
| Lung cancer (C33-4) | 1.71 | (0.51–5.79) | 2.03 | (0.71–5.83) | 1.70 | (0.60–4.81) |

HR: Hazard ratio, CI: Confidence interval, -: If there were no cases in the Jang-jeom village in the period, the calculation is impossible and the HRs and 95% CIs are marked with "-".

* adjusted by age, sex, and income.

## Measurement of PAHs and TSNAs in the collected samples

We confirmed through the Allbaro system that 2,242.0 tons of tobacco leaves were imported and used from 2009 to 2015. However, the imported tobacco leaves that the residents had kept in the village were not registered in the system.

In the samples of deposited dust from the factory, the median values were 2,410.25 µg/kg (range: 175.0–18,871.5 µg/kg) for total PAHs, 9.4 µg/kg (0.7–114.6 µg/kg) for NNN, 36.6 µg/kg (1.3–148.5 µg/kg) for NAT, 1.5 µg/kg (0.1–18.6 µg/kg) for NAB, and 7.3 µg/kg (0.6–38.5 µg/kg) for NNK. More detailed site-specific measurements are described elsewhere [6].

In the residual fertilizer sample collected from inside the stirrer, the average values were 188.9 µg/kg (159.4–218.4 µg/kg) for total PAHs, 6.3 µg/kg (2.7–9.9 µg/kg) for NNN, 30.8 µg/kg (12.8–48.7 µg/kg) for NAT, 1.6 µg/kg (0.6–2.6 µg/kg) for NAB, and 7.1 µg/kg (2.8–11.4 µg/kg) for NNK. In the residual fertilizer sample collected from inside the dryer, the average values were 933.3 µg/kg (180.7–1,686.0 µg/kg) for total PAHs, 19.6 µg/kg (12.3–26.9 µg/kg) for NNN, 62.2 µg/kg (49.8–74.6 µg/kg) for NAT, 3.0 µg/kg (2.5–3.6 µg/kg) for NAB, and 4.3 µg/kg (3.0–5.5 µg/kg) for NNK. High concentrations of total PAHs (196.8 µg/kg), NNN (3,260 µg/kg), NAT (2,310 µg/kg), NAB (94.2 µg/kg), and NNK (1,090 µg/kg) were measured in imported tobacco leaves that were suspected to have been smuggled [6].

The wastewater samples collected from the wastewater collection tank and the raw material mixing section at the plant had high concentrations of total PAHs at 5,932.1 ng/L and 6,471.2 ng/L, respectively. None of the four TSNAs were detected.

In the outdoor dust samples collected in Jang-jeom and the control area, the total PAHs were 174.4 µg/kg (82.8–680.5 µg/kg) and 200.1 µg/kg (54.0–2,736.2 µg/kg), respectively. A small amount of TSNAs was detected in 5 out of 15 spots in Jang-jeom (average: 0.039 µg/kg of NNN, 0.005 µg/kg of NAT, and 0.007 µg/kg of NNK), but no trace of the carcinogens was found in the 5 spots in the control area. More detailed site-specific measurements are described elsewhere [6].

## Discussion

Summarizing the results of the study, 23 cancer cases occurred in Jang-jeom from 2001 to 2017. Between 2010 and 2016, the incidence rates of all cancers (SIR: 2.05, except thyroid

cancer: 2.22), non-melanoma skin cancer (SIR: 21.14, female: 25.41), and GB and biliary tract cancer in men (SIR: 16.01) in the village were higher than those in the national population, with the differences being statistically significant. When comparing SIRs by year (2001–2016) with national population, we found that the SIRs was not high before 2008 and the point estimates has increased since then. Considering the carcinogenetic latency, the temporal trend could be seen as supporting the hypothesis that the pollutants from the plant affected the cancer incidence in the village.

In the cohort analysis that included only Hamra-myeon residents who have lived there for more than 7 years, we found a statistically significant increase in the risk of all cancers (HR: 1.99, except thyroid cancer: 2.20), non-melanoma skin cancer (HR: 11.60), GB and biliary tract cancer (HR: 15.24), liver cancer (HR: 6.63), and gastric cancer (HR: 3.29) for Jang-jeom village residents compared to other Hamra-myeon residents.

In the process of investigating the source of harmful substances, this study found that the fertilizer plant illegally used tobacco leaves for fertilizer production. The levels of PAHs and TSNAs generated in the process of drying tobacco leaves are high, and these pollutants were detected inside the plant. It was also confirmed that TSNAs had already been generated and were present in the tobacco leaves when they were naturally air-dried or artificially heat-dried. We detected small amounts of TSNAs in the dust samples from Jang-jeom, but not in the samples from the control areas, even though the investigation was performed more than a year after the plant had been shut down. It can be reasonably assumed that the residents had at the time been exposed to high concentrations of TSNAs or other harmful pollutants. This assumption is supported by the fact that the plant had no dust collector on the fertilizer dryer was exposed in March, 2017 [1].

The International Agency for Research on Cancer (IARC) has classified benzo[a]pyrene, a type of PAH, and the TSNAs NNN and NNK, some of the most potent carcinogenic substances in tobacco and cigarette smoke, as group 1 human carcinogens [7]. In the case of occupational exposure to PAH, the pooled relative risk of lung cancer was reported to be 1.13 to 1.55 in a meta-analysis [8], and that of bladder cancer was reported to be 1.28 to 1.38 [9]. There are no human epidemiological studies of benzo[a]pyrene, but exposure-related tumors of the lung, trachea, larynx, stomach, liver, lymphoma, skin, and mammary gland have been reported in animal experimental studies [10]. Animal experimental studies have also reported a link between exposure to NNN or NNK and tumors of the lung, nasal cavity, oral cavity, trachea, esophagus, stomach, pancreas, liver, adrenal gland, and skin [11].

According to 2017 cancer registration statistics from the KCCR, there were 232,255 cancer cases in South Korea. The incidence of stomach cancer was 33.3 per 100,000 people, taking the 1st place among all cases; liver cancer incidence was 17.0 per 100,000 and at the 6th place; and GB and biliary tract cancer incidence was 6.7 per 100,000 (2.94% of total cancers) and at the 9th place. Non-melanoma skin cancer accounted for 2.47% of all cancers, with an incidence rate of 5.7 per 100,000 people, which is unlike the high rates in the United States and Europe [12]. Remarkably, our study found high SIRs and HRs of GB and biliary cancer and non-melanoma skin cancer, which, according to available statistics, are rare cancers nationwide. Previous research on the environmental risk factors of GB cancer is limited, but there have been reports that smoking and exposure to heavy metals and radon raise the risk [13]. As regards skin cancer, prospective studies of two big cohorts in the United States showed that the association between smoking and melanoma is reversed, but smoking increases the risk of basal cell carcinoma and squamous cell carcinoma [14].

The reasons why cancer cluster investigations are challenging are that finding actual geographic clusters is difficult, the annual incidence of cancer may be unstable in areas with low population density, and migration of residents may dilute the actual health effects of long-

term exposure to suspected pollutants. Even if the ICD codes in medical records remain the same, the etiology of tumor progression may be different if the molecular types are [15]. The Guidelines for Investigating Clusters of Health Events, published by the US Centers for Disease Control and Prevention, state that a detailed and comprehensive epidemiological investigation would be required when the suspected cluster is for one cancer type or a rare cancer type, cancer occurs at an age when it usually does not, relevant time has elapsed since exposure, and there is guaranteed biological plausibility of the purported association [16]. In our study, the suspected cluster was not about one type of cancer, but about rare cancers, such as GB and biliary cancer and non-melanoma skin cancer, that occurred among the village residents. Moreover, as per our statistical models, the longer the period of residence in the village, the higher the risk (Table 3).

Two issues related to validity may arise because cohort analysis was performed using NHIS database, which is the claims data, rather than the data obtained through direct survey. One is unmeasured confounding issues in statistical models. We controlled the effects of age, sex, and income levels that could be obtained from NHIS database in the Cox model, but we could not control the effects of other potential confounding variables such as smoking and alcohol consumption. However, the group included in the cohort analysis is a population group confined to a small area in Hamra-myeon (22.8 km$^2$), and therefore could be considered to be relatively homogeneous in terms of lifestyle and other environmental circumstances. We believe that we could assess the personal variables about lifestyle if we merge the NHIS health examination database with our cohort datasets in future analyses. Nevertheless, the effect size derived from the Cox model was similar to the effect size from the SIR calculations, increasing the reliability of the results.

The other is the accuracy of cancer diagnosis. In general, the accuracy of diagnosis in insurance billing data tends to be poor. However, in the case of cancer, the diagnosis code is more accurate than other diseases because many tests performed during the diagnosis process are billed together. Moreover, we checked the exemption calculation code for cancer patients, so the probability of including wrong cases or excluding true cases was extremely low.

We also revealed that there is uncertainty in the SIRs for the 2001 to 2009 period. We used the age structure for 2001 to 2009 by reducing the age group by 5 years in the 2009 age structure, as there was no available demographic data of the village from 2001 to 2008. If we had used the same age structure for the 2001 to 2008 period as that in 2009, the SIRs for that period would have been underestimated Furthermore, like other rural areas in South Korea, Jangjeom is also likely to have witnessed a gradual decline in population over time. Thus, if the population of the village in 2001 to 2008 had been higher than in 2009, the actual SIRs in that period would be lower than our estimates. Therefore, it would be reasonable to say that the actual SIRs are likely to be smaller than the estimates presented in S1 Table. In other words, the difference in SIRs for both periods (2001 to 2009 and 2010 to 2016) might be greater in reality, which could be suggestive of a possible effect of long-term exposure due to residence in the village if we consider the latent period of cancer progression.

The inability to assess individual exposure to carcinogens is also one of the shortcomings of our study. Since this investigation was carried out more than a year after the plant had been closed, it was not possible to accurately estimate individual exposure. Because of the short half-life of PAHs and TSNAs, urine or blood biomarkers could not have been detected. Our best option was to measure environmental dust, residual fertilizer, or the raw tobacco leaves. Although PAH and TSNA levels were detected higher in the affected area than in the control area, it was estimated that this elevated concentration level did not generally reach carcinogenic risk [1].

Many experts have noted the importance of reconstructing residential history and improving the utilization of electronic data sources to overcome the limitations of previous cancer cluster investigations [15]. In South Korea, it can be said that accessibility to data sources is good as researchers or investigators can obtain cancer incidence data from certain workplaces or regions and even personal information related to medical use (hospitalization, outpatient, national health examination measurements, region, income level, etc.) from NHIS because all Koreans are insured and NHIS maintains medical treatment databases. This data can be used for the purpose of conducting research with the consent of the subjects. Our study was able to provide strong evidence by constructing longitudinal data using personal medical treatment information and by effectively reconstructing residence histories that were not significantly affected by the effect of dilution due to migration. This is a strong advantage that offsets the above-mentioned shortcomings of this study.

## Conclusion

The results of the SIR calculation and cancer risk analyses of Jang-jeom village residents from the retrospective cohort design showed consistency in the effect size and direction, suggesting that there was a cancer cluster in the village. In particular, the statistical models showed that the longer the period of residence in the village, the higher the risk of cancer. Long-term exposure to pollutants from the fertilizer plant was strongly suspected to be the culprit for the cancer cluster. Long-term exposure to substances from tobacco leaves such as PAH and TSNAs would also guarantee the biological plausibility of cancer development. In addition to PAH and TSNA exposure, the possibility of additional carcinogenic effects from more than 5,300 unidentified harmful substances and other carcinogens in tobacco also needs to be considered. However, it cannot be concluded that the evidence of the association between pollutant exposure from the fertilizer plant and the increased risk of cancer among residents is strong enough or causal. This study would be a good precedent for cancer cluster investigation. Future research could identify possible cancer clusters in an unbiased manner by strengthening the ability to utilize accessible national databases, which we used in this study.

## Supporting information

**S1 Appendix. Sample analysis procedural methods for polycyclic aromatic hydrocarbons (PAHs) and tobacco-specific nitrosamines (TSNAs).**
(DOCX)

**S1 Table. The cancer standardized incidence ratio of the Jang-jeom village according to target population and period.** SIR: Standardized incidence ratio. CI: Confidence interval. -: If there were no cases in the Jang-jeom village in the period, the calculation was impossible and the SIRs and 95% CIs are marked with "-".
(DOCX)

**S2 Table. The hazard ratio of the living in the Jang-jeom village compared to the neighborhood area by the unadjusted model.** HR: Hazard ration. CI: Confidence interval. *: Any covariate is not adjusted. -: If there were no cases in the Jang-jeom village in the period, the calculation was impossible and the SIRs and 95% CIs are marked with "-".
(DOCX)

## Acknowledgments

We would like to thank Editage (www.editage.co.kr) for English language editing.

## Author Contributions

**Conceptualization:** Sanghyuk Bae, Jeong-Soo Kim, Ho-Jang Kwon.

**Formal analysis:** Yong-Han Lee.

**Investigation:** Yong-Han Lee, Sanghyuk Bae, Do-Hyun Koh.

**Methodology:** Yong-Han Lee, Sanghyuk Bae.

**Project administration:** Mira Yoon, Bo-Eun Lee.

**Supervision:** Jeong-Soo Kim, Ho-Jang Kwon.

**Validation:** Do-Hyun Koh.

**Writing – original draft:** Hyungryul Lim.

**Writing – review & editing:** Hyungryul Lim.

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
