## [Decision Letter · Decision Letter 0]

22 Oct 2020

PONE-D-20-23423

Cancer Cluster Among Small Village Residents Near the Fertilizer Plant in Korea

PLOS ONE

Dear Dr. Kwon,

Thank you for submitting your manuscript to PLOS ONE. After careful consideration, we feel that it has merit but does not fully meet PLOS ONE’s publication criteria as it currently stands. Therefore, we invite you to submit a revised version of the manuscript that addresses the points raised during the review process.

We look forward to receiving your revised manuscript.

Kind regards,

Xiaohui Xu, PhD

Academic Editor

PLOS ONE

Journal Requirements:

2. Please provide additional details regarding participant consent.

In the ethics statement in the Methods and online submission information, please ensure that you have specified what type you obtained (for instance, written or verbal, and if verbal, how it was documented and witnessed).

If your study included minors, state whether you obtained consent from parents or guardians.

If the need for consent was waived by the ethics committee, please include this information.

Reviewers' comments:

Reviewer's Responses to Questions

**Comments to the Author**

1. Is the manuscript technically sound, and do the data support the conclusions?

Reviewer #1: Yes

Reviewer #2: Yes

2. Has the statistical analysis been performed appropriately and rigorously? 

Reviewer #1: Yes

Reviewer #2: Yes

3. Have the authors made all data underlying the findings in their manuscript fully available?

Reviewer #1: No

Reviewer #2: No

4. Is the manuscript presented in an intelligible fashion and written in standard English?

Reviewer #1: Yes

Reviewer #2: Yes

5. Review Comments to the Author

Reviewer #1: Lim et al. analyzed data from the National Health Insurance Service in South Korea and identified a local cancer cluster of 23 cancer cases that occurred from 2001 to 2017 in a village. They found that the incidence rates of all cancers, non-melanoma skin cancer, gallbladder, and biliary tract cancer were significantly higher than the general population. Furthermore, they detected known carcinogens including polycyclic aromatic hydrocarbon and tobacco-specific nitrosamines in dust samples collected near a local fertilizer plan in the same village. The manuscript is well-organized and well-written and could be acceptable for publication after minor revisions.

Specific comments.

1. Since there are no line numbers displayed, the following comments are provided based on Introduction, Materials and Methods, Results, and Discussion.

2. Introduction. First paragraph. Please add references (e.g. report, media coverage) because most of the reported data were not collected or directly observed by the authors.

3. Materials and Methods. Collecting National Data. It would be helpful to draw a flow chart to display how data from different sources were integrated and used to identify cancer cases.

4. Materials and Methods. Estimating Cancer Hazard Ratios from Retrospective Cohort Design. Please clarify if everyone in the village enrolled in the NHIS insurance plan. There will be concerns about the misclassification of cases if not everyone in the village participated in the same program.

5. Materials and Methods. Measurement of Polycyclic Aromatic Hydrocarbons and Tobacco-Specific Nitrosamines in Collected Samples. Please add references for the statement “We decided to measure polycyclic aromatic hydrocarbons (PAHs) and tobacco-specific nitrosamines (TSNAs), which are known human carcinogens generated from the processing of tobacco leaves”.

6. Please provide quality control procedures and data for chemical measurements.

7. S2 Table should be shown as part of the main results.

8. Discussion. Sixth paragraph. Please comment on which type of bias would be introduced if using population-level data in the analysis.

9. The statement “but there may have been limitations due to the use of secondary data, such as errors in the process of assigning disease codes. However, codes for cancer cases in hospitals can generally be considered to have high accuracy. Moreover, we checked the exemption calculation code for cancer patients, so the probability of including wrong cases or excluding true cases was extremely low” is self-contradictory. Please consider revising.

Reviewer #2: The paper by Lim et al. investigated a suspected cancer cluster in Jang-jeom, a small village near a fertilize plant that had been using tobacco leaves to produce fertilizers and releasing wastewater close to this village. By using multiple national data sources, the authors were able to compare the cancer incidence in this specific village to that in the general population during affected years, measure and compare concentrations of several chemicals in the dust samples in the affected village versus control areas. Overall, this manuscript is well-written. Some minor problems may need to be addressed.

1. Would it be possible for the authors to show the geographic locations of the plant, the affected village, and the control area (by a map)? From the text it is a bit difficult to tell the geographic relationship among the three.

2. Although PAH and TSNA levels were detected higher in the affected area versus in the control area, does this elevated concentration level/dose reach certain limit that might pose people live nearby at a higher risk for tumors? Please explain.

3. The authors compared the cancer incidence in the affected areas to the national average, what about the affected years versus other years? It is also possible that the cancer incidence in this area is higher than the other places, not only between 2001-2017 when the plant was actively operating, but also other years (which means there might be other reasons that could explain the elevated cancer incidence in this place). Please discuss.

6. PLOS authors have the option to publish the peer review history of their article (what does this mean?). If published, this will include your full peer review and any attached files.

Reviewer #1: No

Reviewer #2: No

---

## [Author Response · Author response to Decision Letter 0]

4 Jan 2021

○ We thank all the reviewers for their comments. We have revised the paper to address all of the issues they noted, and hope they will find our changes to be appropriate and relevant. 

Reviewer #1: Lim et al. analyzed data from the National Health Insurance Service in South Korea and identified a local cancer cluster of 23 cancer cases that occurred from 2001 to 2017 in a village. They found that the incidence rates of all cancers, non-melanoma skin cancer, gallbladder, and biliary tract cancer were significantly higher than the general population. Furthermore, they detected known carcinogens including polycyclic aromatic hydrocarbon and tobacco-specific nitrosamines in dust samples collected near a local fertilizer plan in the same village. The manuscript is well-organized and well-written and could be acceptable for publication after minor revisions.

Specific comments.

1. Since there are no line numbers displayed, the following comments are provided based on Introduction, Materials and Methods, Results, and Discussion.

○ We have added line numbers in the resubmitted manuscript.

2. Introduction. First paragraph. Please add references (e.g. report, media coverage) because most of the reported data were not collected or directly observed by the authors.

○ We have added reference.

3. Materials and Methods. Collecting National Data. It would be helpful to draw a flow chart to display how data from different sources were integrated and used to identify cancer cases.

○ We have replaced table 1. that explained the types and ranges of the data we collected with figure 1., which would be more helpful to understand the overall process. 

4. Materials and Methods. Estimating Cancer Hazard Ratios from Retrospective Cohort Design. Please clarify if everyone in the village enrolled in the NHIS insurance plan. There will be concerns about the misclassification of cases if not everyone in the village participated in the sdname program.

○ We have added description of Korean NHIS in the same paragraph. Because it is mandatory for all citizen (and also Korea has only one insurer, the NHIS), everyone in the village enrolled in the NHIS. 

Materials and Methods > Estimating Cancer Hazard Ratios … > 1st paragraph

In South Korea, the whole population is enrolled in one central insurer, NHIS. Since NHIS enrollment is mandatory, through the insurance system, we could obtain the qualification database of the service (income level, resident area, etc.) and medical treatment databases (date of visit, ICD code, prescription, etc.) of all Hamra-myeon residents including Jang-jeom villagers and reconstruct a detailed cohort at an individual level.

5. Materials and Methods. Measurement of Polycyclic Aromatic Hydrocarbons and Tobacco-Specific Nitrosamines in Collected Samples. Please add references for the statement “We decided to measure polycyclic aromatic hydrocarbons (PAHs) and tobacco-specific nitrosamines (TSNAs), which are known human carcinogens generated from the processing of tobacco leaves”.

○ We have cited IARC report in the text.

6. Please provide quality control procedures and data for chemical measurements.

○ We have made a supplementary document for detailed procedural method.

7. S2 Table should be shown as part of the main results.

○ We have replaced S2 Table (demographic characteristics) to Table 1. in main body.

8. Discussion. Sixth paragraph. Please comment on which type of bias would be introduced if using population-level data in the analysis.

○ As you mentioned, we used National Health Insurance Service database that ifself is for insurance coverage billing rather than investigating epidemiologic study. So, there were two inevitable limitations. 

One is unmeasured confounding issues in statistical models. We’ve adjusted age, sex, and household income in cancer risk models. We cannot help to admit lacking possible confounder in our models, for example smoking, and alcohol consumption, due to database limitation. But study area (including control area) is pretty small (22.8 km2), so therefore the study population could be considered as homogenous in terms of lifestyle and environmental circumstances.

The other is the accuracy of cancer diagnosis. In general, NHIS database is for billing, so the accuracy of diagnosis tends to be poor. However, in the case of cancer, the diagnosis code is more accurate than other diseases. Moreover, we checked the exemption calculation code for cancer patients, so the probability of including wrong cases or missing true cases was extremely low.

We have revised this in Discussion, 7th-8th paragraph as follows:

Two issues related to validity may arise because cohort analysis was performed using NHIS database, which is the claim data, rather than the data obtained through direct. One is unmeasured confounding issues in statistical models. We controlled the effects of age, sex, and income levels that could be obtained from NHIS database in the Cox model, but we could not control the effects of other potential confounding variables such as smoking and alcohol consumption. However, the group included in the cohort analysis is a population group confined to a small area in Hamra-myeon (22.8 km2), and therefore could be considered to be relatively homogeneous in terms of lifestyle and other environmental circumstances. We believe that we could assess the personal variables about lifestyle if we merge the NHIS health examination database with our cohort datasets in future analyses. Nevertheless, the effect size derived from the Cox model was similar to the effect size from the SIR calculations, increasing the reliability of the results.

The other is the accuracy of cancer diagnosis. In general, the accuracy of diagnosis in insurance billing data tends to be poor. However, in the case of cancer, the diagnosis code is more accurate than other diseases because many tests performed during the diagnosis process are billed together. Moreover, we checked the exemption calculation code for cancer patients, so the probability of including wrong cases or excluding true cases was extremely low.

9. The statement “but there may have been limitations due to the use of secondary data, such as errors in the process of assigning disease codes. However, codes for cancer cases in hospitals can generally be considered to have high accuracy. Moreover, we checked the exemption calculation code for cancer patients, so the probability of including wrong cases or excluding true cases was extremely low” is self-contradictory. Please consider revising.

○ As you pointed out, this paragraph seems to be self-contradictory, we have revised this paragraph (Discussion, 8th paragraph) with more detailed explanation as follows.

The other is the accuracy of cancer diagnosis. In general, the accuracy of diagnosis in insurance billing data tends to be poor. However, in the case of cancer, the diagnosis code is more accurate than other diseases because many tests performed during the diagnosis process are billed together. Moreover, we checked the exemption calculation code for cancer patients, so the probability of including wrong cases or excluding true cases was extremely low.

------------------

Reviewer #2: The paper by Lim et al. investigated a suspected cancer cluster in Jang-jeom, a small village near a fertilize plant that had been using tobacco leaves to produce fertilizers and releasing wastewater close to this village. By using multiple national data sources, the authors were able to compare the cancer incidence in this specific village to that in the general population during affected years, measure and compare concentrations of several chemicals in the dust samples in the affected village versus control areas. Overall, this manuscript is well-written. Some minor problems may need to be addressed.

1. Would it be possible for the authors to show the geographic locations of the plant, the affected village, and the control area (by a map)? From the text it is a bit difficult to tell the geographic relationship among the three.

○ We have added a regional map (Fig. 2)

2. Although PAH and TSNA levels were detected higher in the affected area versus in the control area, does this elevated concentration level/dose reach certain limit that might pose people live nearby at a higher risk for tumors? Please explain.

○ In the risk assessment conducted by government researchers, it was estimated that the level of PAHs and TSNA did not generally reach carcinogenic risk. 

We have additionally added the issues in Discussion > 10th paragraph, as follows:

Although PAH and TSNA levels were detected higher in the affected area than in the control area, it was estimated that this elevated concentration level did not generally reach carcinogenic risk [1].

For these reasons, we mentioned in conclusion section that 

‘it cannot be concluded that the evidence of the association between pollutant exposure from the fertilizer plant and the increased risk of cancer among residents is strong enough or causal.’

3. The authors compared the cancer incidence in the affected areas to the national average, what about the affected years versus other years? It is also possible that the cancer incidence in this area is higher than the other places, not only between 2001-2017 when the plant was actively operating, but also other years (which means there might be other reasons that could explain the elevated cancer incidence in this place). Please discuss.

○ Unfortunately NHIS database has been provided since 2002, we could not estimate HRs for the period before the plant in operation. That could be a limitation of the study. However, when comparing SIRs by year (2001-2016) with national population, we found that the SIRs was not high before 2008 and the point estimates has increased since then. 

Furthermore, Hamra-myeon, the study area, is quite small area of 22.8km2, the demographic characteristics and environment of the both control area and Jang-jeom village are almost similar. (mentioned in Discussion) Considering this homogeneity and the carcinogenetic latency, it could be seen that pollutants from the plant were likely to increase the risk of cancer. 

We have additionally added the issues in Discussion > 1st paragraph, as follows:

When comparing SIRs by year (2001-2016) with national population, we found that the SIRs was not high before 2008 and the point estimates has increased since then. Considering the carcinogenetic latency, the temporal trend could be seen as supporting the hypothesis that the pollutants from the plant affected the cancer incidence in the village.

---

## [Editor Report · Decision Letter 1]

11 Feb 2021

Cancer Cluster Among Small Village Residents Near the Fertilizer Plant in Korea

PONE-D-20-23423R1

Dear Dr. Kwon,

We’re pleased to inform you that your manuscript has been judged scientifically suitable for publication and will be formally accepted for publication once it meets all outstanding technical requirements.

Kind regards,

Xiaohui Xu, PhD

Academic Editor

PLOS ONE
---

## [Editor Report · Acceptance letter]

16 Feb 2021

PONE-D-20-23423R1 

Cancer Cluster Among Small Village Residents Near the Fertilizer Plant in Korea 

Dear Dr. Kwon:

I'm pleased to inform you that your manuscript has been deemed suitable for publication in PLOS ONE. Congratulations! Your manuscript is now with our production department. 

Kind regards, 

on behalf of

Dr. Xiaohui Xu 

Academic Editor

PLOS ONE